# Dose blockchain-based agri-food supply chain guarantee the initial information authenticity? An evolutionary game perspective

**Weixia Yang[1], Congli Xie[1], Lindong Ma[2,3]***

**1** School of Business, Xi'an International University, Xi'an, China, **2** Xingzhi College, Zhejiang Normal University, Jinhua, China, **3** School of Management, Zhejiang University of Technology, Hangzhou, China

* malindong@zjnu.edu.cn

## Abstract

Guarantee the initial information of the agri-food supply chain (AFSC) authenticity based on the blockchain is a complex problem. This paper develops an evolutionary game model of AFSC participants based on the blockchain and discusses the impacts of the key parameters on the dynamic evolution process of participants. To verify the theoretical results, simulation experiments and sensitivity analysis were conducted through *Matlab 2022b*. The study results show that: (1) Guaranteeing the initial information authenticity could become the common belief of all AFSC participants, with the scientific design of parameters; (2) Higher reward and synergistic effect, lower information cost and risk contribute to improving the probability of initial true information sharing. (3) when the default penalty is too severe, the enterprise will evolve into not sharing the initial true information. Finally, this study could provide some suggestions and countermeasures for the leading enterprise in the agricultural supply chain and local governments to guarantee initial information authenticity in China. That is the way to realize the sustainability of AFSC in the long run.

**Data Availability Statement:** All relevant data are available at: 10.6084/m9.figshare.22655179.

**Funding:** This study was funded by The General soft science project of Shaanxi Science and Technology Department (grant number

## 1. Introduction

Agriculture plays a crucial fundamental role in China's national economy. Its annual production of agri-food ranks first in the world, and the gross output value of agriculture, forestry, animal husbandry, and fisheries continues to grow yearly. agri-food are closely related to people's lives. To meet the needs of the people, the government and enterprises have been exploring ways to ensure the quality and safety of agricultural products. The AFSC is a supply chain network system composed of upstream and downstream nodes, including farmers, agri-food traders, purchasing and processing enterprises, distribution, retailers, logistics distributors, and final consumers [1, 2]. Mutual trust, deep cooperation, and shared interests and risks among the members of the AFSC have become the key to improving the chain's value-added due to the complexity of the main body composition and production relations in the AFSC [3, 4], the importance of product quality and safety, the limitation of time competition, and the difficulty of logistics management [5]. Currently, in China, the drawbacks of the AFSC are

2022KRM137) awarded to WY and by Xi'an Social Science Fund General Project (grant number 22JX152) awarded to WY. The funders had no role in study design, data collection and analysis, decision to publish, or preparation of the manuscript.

**Competing interests:** The authors have declared that no competing interests exist.

gradually revealing themselves due to the scattered and numerous participants and changes in the external environment [6]. For example, the supply chain information transparency is low for agri-food, the power asymmetry among participants lacks trust, the tendency of opportunism in the supply chain is self-evident, and the security in transactions is lacking [7, 8]. These circumstances increase the vulnerability of the AFSC and lead to a supply-chain failure rate of more than 50% [9]. The phenomenon has attracted the attention of scholars who regard information asymmetry as a fundamental reason for the intensified conflict and hindered quality and efficiency improvement in the AFSC [10, 11].

As a disruptive integrated innovation technology, blockchain is an open and shared database formed by computing each subnode through cryptography and using the Internet as a carrier [12, 13]. The era of digitally driven agriculture led by blockchain will shortly begin, and data uploading has become one of the important trends in the future development of the AFSC [14, 15]. There are multiple couplings between decentralization, high openness, anonymity, machine autonomy, information immutability and traceability of the blockchain, and the above-mentioned pain points of the agricultural supply chain operations. (1) As a decentralized data storage technology, blockchain can integrate the decentralized node members of the AFSC into an overall block information chain [16]. Therefore, each participant in the supply chain can interact to share the ledger and realize the transparency and reliability of the information on the chain, which can effectively solve the problem of information asymmetry in the traditional AFSC [11, 17]. (2) The blockchain protects data based on the principles of cryptography. Only when private keys and public keys are used together along with more than 51% of the nodes reaching a consensus can the authenticity of the data be confirmed [18, 19]. A relationship of mutual trust can be established among the nodes of the agricultural supply chain, and the information barriers can be reduced, creating a relationship of mutual trust and reducing information barriers [20]. (3) Each block consists of a header, body, and timestamp. The data on the chain cannot be illegally tampered with, and any data can be accurately traced to their origin along the chain [21, 22]. This method solves problems such as data tampering, deceiving fake goods, and shirking reliability can be prevented. Additionally, food safety can be tracked throughout the whole process, and throughout the entire AFSC [23]. (4) When the data on the blockchain meets certain conditions, the smart contract can automatically verify and execute the pre-agreed contract, enabling trusted transactions and asset transfers without third-party authorization [24]. This decentralized application eliminates intermediaries and supply-chain control centers, enhances the symmetry of the supply-chain structure, changes the current situation in which the chain is strong in the middle and weak at both ends, and improves the efficiency of the agricultural supply chain.

In short, blockchain technology is digital technology that has begun changing the relationship between the AFSC and reshaping the ecosystem of agri-food. Applying blockchain to agri-food circulation can bridge the gap between various stakeholders in agri-food circulation, resolve conflicts of interest, and rebuild the trust system and interest pattern [25, 26]. In 2020, the central "No. 1 Document" also proposed to speed up the application of blockchain in the agricultural field and promote the application of blockchain in the AFSC in China. Currently, the blockchain technology embedded in the AFSC operation cannot ensure the authenticity of the initial information. Parts of the chain will still protect their interests by lying about the costs for additional benefits, making the authenticity verification and supervision of the initial information in the blockchain technology expensive [27, 28].

One suggestion to solve this problem involves playing a game with the participants in the AFSC based on the blockchain [29, 30]. Each node can then be analyzed to determine whether it follows the protocol to ensure the initial real information sharing after entering the blockchain, thereby improving the AFSC [31, 32]. This evolutionary game is a repeated game based

on bounded rationality. Since the game participants are bounded by rationality, they adjust their strategies through continuous learning and imitation of the optimal strategy and form a stable strategy, allowing the game to reach an equilibrium state. We refer to this strategy as the "evolutionary stabilization strategy" [33]. The realization of this equilibrium state is a process of continuous adjustment and optimization, which has a certain level of stability and can autonomously return to a stable state after being biased by a small amount of disturbance [34, 35]. In this paper, the information sharing of the AFSC based on blockchain is essentially a contractual partnership based on agri-food-related information. Due to the lack of definite and complete decision-making information between the supply and demand sides of the supply chain, it is uncertain whether to follow the protocol to ensure that the initial real information is shared after entering the blockchain. This process is subject to various influencing factors [36]. The repeated game gradually forms a sound rationale and strategy.

Therefore, the key approach in this paper starts with the selection strategy set of parties and uses the evolutionary game model to explore whether the parties follow the agreement to ensure the initial real information enters the blockchain [37]. The goals are to improve the AFSC, study the strategies of parties when they reach a steady state and the conditions under which they hold, and give corresponding suggestions for different situations.

The remainder of the paper is organized as follows: Section 2 reviews related research. Section 3 describes the issue and its corresponding assumption, and it builds the framework for the basic model. Evolutionary analyses among the game players are also presented. Moreover, details of methods, including the establishment of the game model, replicator dynamic, and ESS analysis of each agent, along with the definition of an ideal event and its determining factors are presented in Section 3. Section 4 discusses the sensitivity analysis of selected parameters in detail. Finally, Section 5 provides summary conclusions and managerial implications.

## 2. Literature review

In the agri-food supply chain, the blockchain has become an important element for guaranteeing the quality and safety of agricultural products in the market, and collaboration among participants is the key to improving the core competitiveness of the supply chain. In this context, many scholars have carried out relevant research. Therefore, the literature review is conducted from two aspects as follows:

### 2.1 Application of blockchain in the field of AFSC

Currently, academic research on the application of blockchain in the field of AFSCs focuses on application scenarios and effects.

**2.1.1 Study on application scenarios of blockchain technology in the operation of the AFSC.** Ronaghi, M.H.(2020) [38] constructed a hierarchical structure model of "blockchain + AFSC" to realize information sharing and in-depth cooperation, such that the information optimization of the AFSC was achieved. Yu, C. et al. (2020) [39] built a quality-and-safety-traceability system for agri-food from a blockchain application layer, data layer, core layer, and physical layer architecture to realize the safety traceability of agri-food. Yang, X. et al. (2021) [40] constructed the traceability system of the fruit supply chain based on "blockchain + Internet of Things," and proposed the path optimization of a supply-chain traceability system. Wang, J. et al. (2022) [41] constructed a blockchain-driven information-supervision model for the rice supply chain to ensure the information interconnection of AFSC. These studies covered most aspects of the application of blockchains in the field of agricultural supply chains but lacked systematic and dynamic research methods.

**2.1.2 Research on the improvement of the AFSC operation by embedding blockchain technology.** Behzadi, G. et al. (2017) [42] discussed the industrial chain governance mechanism of agricultural forms in the era of big data and pointed out that opportunism is effectively constrained in the governance of the agricultural-industrial chain based on blockchain. Sun, Y. et al. (2018) [43] proposed a flexible and trusted traceability solution for agri-food based on blockchain. Alkahtani, M.(2021) [37] proposed that blockchain technology can reduce the management costs and operational risks of the AFSC. To improve the stability of the supply chain of fresh agri-food. Niu, B. et al. (2021) [44] used blockchain technology to set pricing and ordering decisions and discourage producers from falsely reporting vivid information. Mukherjee, A.A. et al. (2021) [12] found that blockchain technology could not ensure the authenticity of the initial information of the blockchain. These studies did not analyze the feasibility of the implementation of blockchain in the AFSC. In China, the participants in the supply of agri-food, such as farmers, are scattered and small in scale. Since they lack capital, technology, information, and other aspects, it is difficult for them to integrate effectively into the blockchain [45].

## 2.2 Research on the cooperation modes and game relationships among the participants in the AFSC

Van Ozkan-Canbolat, E. et al. (2016) [46] built a multi-party evolutionary game model to determine whether the participants in the AFSC keep their promises and analyzed the influencing factors behind whether the participants in the AFSC follow the agreement to share information. Ho, K.L.P., et al. (2017), Wang, C. et al. (2017) [47, 48] took the fresh agri-food supply chain as an example to build a game model and discuss decision-making behavior from the perspective of sharing preconditions and conditions, incentive mechanisms, channels, and platforms. The above studies either carried out game analysis from a static point of view or performed the dynamic analysis of information sharing among supply-chain nodes. Shahid, A. et al. (2020) [49] used a collaborative game model to study the stability of internal contractual cooperation in the supply chain of ecological agri-food and analyzed how blockchain technology can guarantee the stable operation of the ecological AFSC. Bergen, M. et al. (2019), and Wu, Y. et al. (2021) [50, 51] built a game model of financial sharing behavior between agri-food suppliers and processors and analyzed the stability of the model to promote the smooth implementation of financial sharing in the AFSCs. However, there is no dynamic evolution analysis on the confidence sharing of the AFSC embedded in blockchain technology [52]. Additionally, there is no stability analysis on the sharing of the initial real information of each node enterprise in the AFSC in the blockchain.

The following elements are the differences between our work and the related literature: (1) The paper intends to construct an evolutionary game model to describe the decision-making process of the initial real information sharing of the AFSC; (2) The research analyzes the influence of each factor and its change on the game process; (3) The study provides useful decision-making tools for enterprises and government.

## 3. Construction of an evolutionary game model

### 3.1 Model assumption

The following hypotheses are advanced according to the needs of the model.

**Hypothesis 1.** The participants in AFSC include farmers, enterprises, wholesalers, retailers, and consumers [53]. To highlight the research focus, the game model simplifies the agricultural supply chain system based on blockchain technology and assumes that the game

players are upstream supplier A and core enterprise B. It can also be any pair of cooperative relationships between farmers and purchasers, cooperatives and purchasers, or purchasers and distributors [54, 55]. The two-game agents have bounded rationality [56].

**Hypothesis 2.** when the information of each node in the blockchain is asymmetric, the two sides of the game choose the strategy based on the maximization of interests [57]. The strategy spaces of players A and B are [1, 2], in which breach and performance denote sharing the initial reliable information as contract keeping and not sharing the initial information or sharing unveracious information [3, 16]. Among them, the probability of player A performance is x ($0 \leq x \leq 1$), and the probability of player A breach is $1-x$. Similarly y ($0 \leq y \leq 1$) and $1-y$ are the probability of player B with a choice performance and breach respectively.

**Hypothesis 3.** $N_i$ is the normal income when both sides of the game do not share real initial information. The two players of the game have different degrees of trust in the other player in sharing real information. The higher the degree of trust, the more real information is shared, and the more income the node enterprise that receives the information will receive. $\theta_i$ ($i$ = a, b) is the amount of initial real information shared by both sides, respectively.

**Hypothesis 4.** When both sides of the game choose a shared strategy, both sides can absorb and utilize real information to gain additional benefits. $L_i$ is the ability of the party to absorb and utilize the initial real information shared by the other party, then $\lambda$ is the synergistic effect coefficient brought to both parties by players A and B, keeping the contract at the same time. Therefore, when player A obeys the contract, the additional income that player B obtains from sharing the initial real information is $\lambda L_b \theta_a$. Similarly, the additional income of player A is $\lambda L_a \theta_b$.

**Hypothesis 5.** When both sides of the game share the initial real information, they receive corresponding incentives. These incentives include the improvement of the image and reputation brought about by the sharing of the initial real information and the material rewards of other node enterprises to the shared members. When sharing increases, so do the incentives. "m" is the compensation incentive coefficient obtained by sharing real information. When player A honors the contract, the reward for A is $m\theta_a$, and when player B honors the contract, the reward for B is $m\theta_b$.

**Hypothesis 6.** When one party shares and the other party does not, a penalty mechanism is introduced to motivate the non-sharing party to share the initial real information [58]. When player A keeps their promise and B defaults, player B is given a punishment $P_b$; Similarly, player A is given a punishment $P_a$. $\lambda$ is the synergistic effect coefficient given to both parties when players A and B keep the contract at the same time.

**Hypothesis 7.** There are certain risks when the two parties share the initial real information [59]. These risks are mainly reflected in the loss of their advantages due to the sharing of real information and the risk of real information being leaked by cooperative enterprises [60]. $J$ is the risk cost coefficient of sharing initial truth information. When player A keeps the contract, then the risk cost borne by A is $J\theta_a$. When player B keeps the contract, then the risk cost borne by B is $J\theta_b$. In addition, Ci is other cost of the initial real information-sharing, cluding the time cost, the material cost of sharing, and the cost of information technology [61]. The definitions of each parameter are listed in **Table 1**.

**Table 1. Parameters' setting.**

| PARAMETERS | DEFINITIONS |
|---|---|
| $X$ | The probability of player A's compliance ($0 \leq x \leq 1$); |
| $Y$ | The probability of player B's compliance ($0 \leq y \leq 1$); |
| $N_I$ | The normal income when both sides of the game do not share real initial information ($i = a,b$); |
| $\Theta_I$ | The amount of initial real information shared by both sides of the game ($i = a,b$); |
| $L_I$ | When one party of the game shares the initial real information, the other party's absorptive capacity for it ($i = a,b$); |
| $M$ | The reward and incentive coefficient obtained when both sides of the game share real information; |
| $\gamma$ | Both sides of the game keep the contract at the same time, which brings synergy to both sides ($0 \leq \gamma \leq 1$); |
| $J$ | The coefficient of the risk cost for both parties to share the initial real information; |
| $C_I$ | Other costs for both parties of the game to share the initial truth information ($i = a,b$); |
| $P_I$ | The penalty payable in the event of default by one party ($i = a,b$) |

## 3.2 Model matrix construction

According to the above hypotheses, the game payoff matrix of the two-party evolutionary game is shown in **Table 2.**

## 3.3 Evolutionary stability analysis

### 3.3.1 Stability analysis of player A.
In deriving the utility functions based on these matrixes, the respective expectation value of "performance" and "breach" player A are $U_{a1}$ and $U_{a2}$. Player A's average value is $U_a$, then:

$$U_{a1} = y(N_a + m\theta_a + \theta_b L_a + \gamma\theta_a - C_a - J\theta_a) + (1-y)(N_a + m\theta_a - C_a - J\theta_a) \qquad (1)$$

$$U_{a2} = y(N_a + \theta_b L_a - P) + (1-y)N_a \qquad (2)$$

$$U_a = x*U_{a1} + (1-x)U_{a2} \qquad (3)$$

According to the Malthusian dynamic equation, the growth of the "performance" strategy adopted by player A should be equal to the expected utility $U_{a1}$, minus the average expected utility $U_a$ [62]. Therefore, the replicator dynamics formula of the "performance" strategy selected by player A is as follows:

$$F(x) = \frac{dx}{dt} = x(U_{a1} - U_a) = x(1-x)(U_{a1} - U_{a2}) = x(1-x)[y(\gamma\theta_a + P) + m\theta_a - C_a - J\theta_a] \qquad (4)$$

According to the stability theorem of differential equations and the property of ESS, the ESS point must be robust to account for minor disturbances. Specifically, when the value of x

**Table 2. Evolutionary game matrix of both players.**

| GAME AGENT AND SELECT STRATEGY | | PLAYER B | |
|---|---|---|---|
| | | Performance(*y*) | Breach(*1-y*) |
| PLAYER A | Performance (*x*) | $N_a + m\theta_a + \theta_b L_a + \gamma\theta_a - C_a - J\theta_a, N_b + m\theta_b + \theta_a L_b + \gamma\theta_b - C_b - J\theta_b$ | $N_a + m\theta_a - C_a - J\theta_a, N_b + \theta_a L_b - P$ |
| | Breach (*1-x*) | $N_a + \theta_b L_a - P,$ $N_b + m\theta_b - C_b - J\theta_b$ | $N_a,$ $N_b$ |

becomes smaller than x*, F(x) must be greater than zero. However, when the value of x becomes larger than x*, F(x) must be smaller than zero. Therefore, $F(x) = 0$ and $F'(x) < 0$ are required to achieve ESS.

The derivative of F(x) is as follows:

$$\frac{dF(x)}{dx} = (1 - 2x)\left[y(\gamma\theta_a + P) + m\theta_a - C_a - J\theta_a\right] \tag{5}$$

Evaluating Eq (4), x =0, x =1 and $y^* = (C_a + J\theta_a - m\theta_a)/(\gamma\theta_a + P)$ are the roots of F(x) =0.

**Proposition 1.**

1. If $y = y^* = (C_a + J\theta_a - m\theta_a)/(\gamma\theta_a + P)$, then for any x, $F(x) = 0$ and $F'(x) = 0$, then axis x is in a stable state, and all game strategies of player A are stable strategies.

2. If $y \neq y^* = (C_a + J\theta_a - m\theta_a)/(\gamma\theta_a + P)$, supposing $F(x) = 0$, then x = 0 and x = 1 are two stable points of x.

**Proof 1.** Then, three circumstances are discussed separately according to Eq (5):

**Case 1.** If $C_a + J\theta_a - m\theta_a < 0$, then $y > (C_a + J\theta_a - m\theta_a)/(\gamma\theta_a + P)$ for two solutions x = 0, x = 1 of Eq (4), when x = 1, $F'(x) < 0$; when x = 0, $F'(x) > 0$, we can see that x = 1 is the only ESS, as shown in **Fig 1(A)**. Bounded by rationale player A will adopt a "performance" strategy.

Case 1 shows that player A can gain more profits by sharing the initial real information than not sharing, after which player A adopts the "performance" strategy anyway.

**Case 2.** If $0 < C_a + J\theta_a - m\theta_a < \gamma\theta_a + P$, then when $y > (C_a + J\theta_a - m\theta_a)/(\gamma\theta_a + P)$, when x = 1, $F'(x) < 0$; when x = 0, $F'(x) > 0$, we can see that x = 1 is the only ESS. When $y < (C_a + J\theta_a - m\theta_a)/(\gamma\theta_a + P)$, when x = 1, $F'(x) > 0$ and when x = 0, $F'(x) < 0$, we can see that x = 0 is the only ESS, as shown in **Fig 1(B)**.

Case 2 shows that player A can gain less profit from sharing the initial real information than not sharing, after which player A adopts the "breach" strategy.

**Case 3.** If $C_a + J\theta_a - m\theta_a > \gamma\theta_a + P$, then $y < (C_a + J\theta_a - m\theta_a)/(\gamma\theta_a + P)$, for two solutions x = 0, x = 1 of Eq (4), when x = 1, $F'(x) > 0$; when x = 0, $F'(x) < 0$; we can see that x = 0 is the only ESS, as shown in **Fig 1(C)**.

Case 3 means that player A can gain more profits from not sharing the initial real information than sharing, after which player A adopts the "breach" strategy anyway.

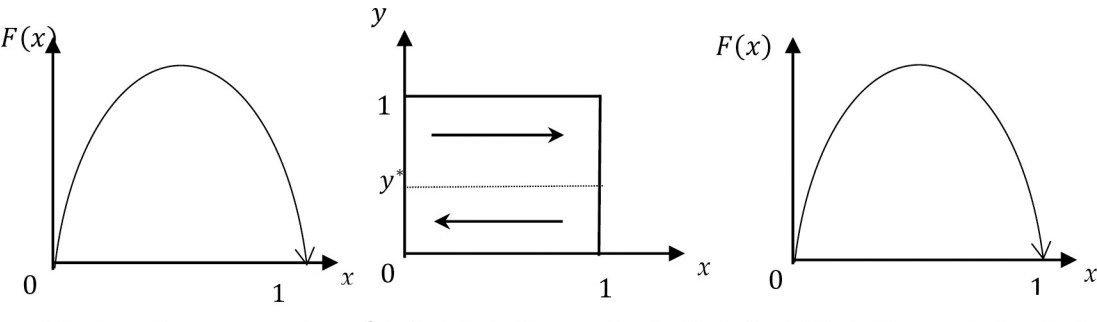

(a) $C_a + J\theta_a - m\theta_a < 0$      (b) $0 < C_a + J\theta_a - m\theta_a < \gamma\theta_a + P$      (c) $C_a + J\theta_a - m\theta_a > \gamma\theta_a +$

**Fig 1. Phase diagrams for player A's strategies.** (a) $C_a + J\theta_a - m\theta_a < 0$, (b) $0 < C_a + J\theta_a - m\theta_a < \gamma\theta_a + P$, and (c) $C_a + J\theta_a - m\theta_a > \gamma\theta_a + P$.

**3.3.2 Strategy stability analysis of player B.**   In the same way, the respective expectation value of "performance" and "breach" player B is $U_{b1}$ and $U_{b2}$. Player B's average value is $U_b$, then:

$$U_{b1} = x(N_b + m\theta_b + \theta_a L_b + \gamma\theta_b - C_b - J\theta_b) + (1-x)(N_b + m\theta_b - C_b - J\theta_b) \qquad (6)$$

$$U_{b2} = x(N_b + \theta_a L_b - P) + (1-x)N_b \qquad (7)$$

$$U_b = y * U_{b1} + (1-y)U_{b2} \qquad (8)$$

The replicator dynamics formula of the "performance" strategy selected by player B is as follows:

$$F(y) = \frac{dy}{dt} = y(U_{b1} - U_b) = y(1-y)(U_{b1} - U_{b2}) = y(1-y)[x(\gamma\theta_b + P) + m\theta_b - C_b - J\theta_b)] \quad (9)$$

The derivative of F(y) is as follows:

$$\frac{dF(y)}{dy} = (1-2y)\left[x(\gamma\theta_b + P) + m\theta_b - C_b - J\theta_b)\right] \qquad (10)$$

It can be easily demonstrated that $y = 0$, $y = 1$ and $x^* = (C_b + J\theta_b - m\theta_b)/(\gamma\theta_b + P)$ are the roots of $F(y) = 0$.

**Proposition 2.**   1.   If $x = x^* = (C_b + J\theta_b - m\theta_b)/(\gamma\theta_b + P)$, $F(y) = 0$ and $F'(y) = 0$, then any regulatory strategies of player B are stable strategies.

2.   If $x \neq x^* = (C_b + J\theta_b - m\theta_b)/(\gamma\theta_b + P)$, supposing (y) = 0, then y = 0 and y = 1 are two stable points of y; we demonstrate the different case as follows:

**Proof 2.** Three circumstances result from Eq (10):

**Case 4.** If $C_b + J\theta_b - m\theta_b < 0$, then $x > (C_b + J\theta_b - m\theta_b)/(\gamma\theta_b + P)$ for two solutions $y = 0$, $y = 1$ of Eq (9), when $y = 1$, $F'(y) < 0$; when $y = 0$, $F'(y) > 0$, we can see that $y = 1$ is the only ESS, as shown in **Fig 2(A)**. Bounded rational player B will adopt the "performance" strategy.

Case 4 shows that player B can gain more profits by sharing the initial real information than by not sharing, and then player B will adopt the "performance" strategy anyway.

**Case 5.** If $0 < C_b + J\theta_b - m\theta_b < \gamma\theta_b + P$, then when $x > (C_b + J\theta_b - m\theta_b)/(\gamma\theta_b + P)$, when $y = 1$, $F'(x) < 0$; when $y = 0$, $F'(y) > 0$, we can see that $y = 1$ is the only ESS. When

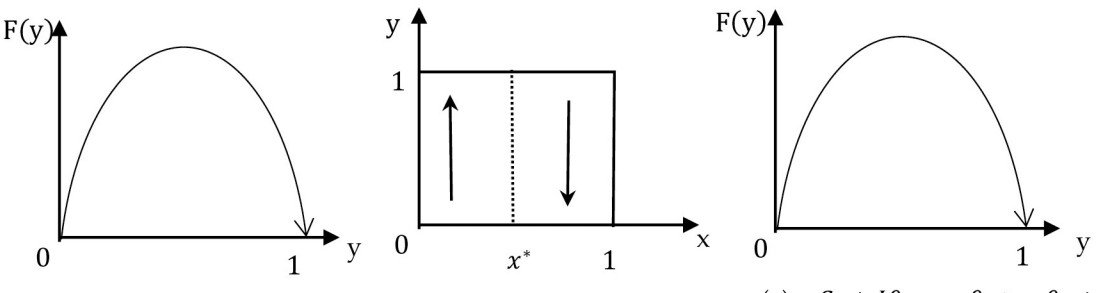

(a)  $C_b + J\theta_b - m\theta_b < 0$     (b)   $0 < C_b + J\theta_b - m\theta_b < \gamma\theta_b + P$     (c)   $C_b + J\theta_b - m\theta_b > \gamma\theta_b + P$

**Fig 2.  Phase diagrams for player B's strategies.** (a) $C_b + J\theta_b - m\theta_b < 0$, (b) $0 < C_b + J\theta_b - m\theta_b < \gamma\theta_b + P$, and (c) $C_b + J\theta_b - m\theta_b > \gamma\theta_b + P$.

$x < (C_b + J\theta_b - m\theta_b)/(\gamma\theta_b + P)$, when y = 1, $F'(y) > 0$ and when y = 0, $F'(y) < 0$, we can see that y = 0 is the only ESS, as shown in **Fig 2(B)**.

Case 5 shows that player B can gain less profit from sharing the initial real information than not sharing, after which player B adopts the "breach" strategy.

**Case 6.** If $C_b + J\theta_b - m\theta_b > \gamma\theta_b + P$, then $y < (C_b + J\theta_b - m\theta_b)/(\gamma\theta_b + P)$, for two solutions y = 0, y = 1 of Eq (9), when y = 1, $F'(y) > 0$ and when y = 0, $F'(y) < 0$, we can see that y = 0 is the only ESS, as shown in **Fig 2(C)**.

Case 6 means that player B can gain more profits from not sharing the initial real information than sharing, after which player B adopts the "breach" strategy anyway.

**3.3.3 Stability analysis between both sides of the game.** Based on the above analysis, players A and B have three stability strategies under different initial conditions. In China's current state, the research and development of blockchain technology are not yet mature, and its application in the AFSC is still in its infancy; therefore, the farmers have limited ability, making it difficult for them to integrate effectively into the operation of the blockchain [11, 44]. Cases 1 and 4 are impossible, and cases 3 and 6 are beyond the topical scope of this paper. Therefore, this section further discusses the dynamic evolution of the system under cases 2 and 5.

Participants constantly adjust their strategies in pursuit of the maximization of their interests, and the strategy that finally achieves a dynamic balance is called the Evolutionary Stability Strategy (ESS). The replicator dynamic system is obtained according to Eqs (4) and (9), which is a two-dimensional nonlinear dynamic system for players *A* and *B*, as follows:

$$F(x) = \frac{dx}{dt} = x(U_{a1} - U_a) = x(1 - x)(U_{a1} - U_{a2}) = x(1 - x)[y(\gamma\theta_a + P) + m\theta_a - C_a - J\theta_a]$$

$$F(y) = \frac{dy}{dt} = y(U_{b1} - U_b) = y(1 - y)(U_{b1} - U_{b2}) = y(1 - y)[x(\gamma\theta_b + P) + m\theta_b - C_b - J\theta_b]$$

**Proposition 3.** Under the condition: $0 < C_a + J\theta_a - m\theta_a < \gamma\theta_a + P$ and $0 < C_b + J\theta_b - m\theta_b < \gamma\theta_b + P$, five replicated dynamic equilibrium points are obtained, A(0,1), B(0,0), C(1,0), D(1,1) and E (x*,y*), if and only if the desired condition x*∈[0,1] and y*∈[0,1] is established.

Where $x^* = (C_b + J\theta_b - m\theta_b)/(\gamma\theta_b + P)$, $y^* = (C_a + J\theta_a - m\theta_a)/(\gamma\theta_a + P)$

For the five equilibrium points of the replicator dynamic system, we can deduce that the equilibrium points A(0,1) and C(1,0) are in an unstable state, B(0,0) and D(1,1) are asymptotic to the system, and E (x*,y*) is a saddle point.

**Proof 3.** Let the eqs F(x) = 0 and F(y) = 0 be used to replicate the dynamic Eqs (4) and (9) [62]. Therefore, when the rate of change of the system strategy selection is zero, five replicated dynamic equilibrium points are obtained: A(0,1), B(0,0), C(1,0), D(1,1), and E (x*,y*).

It is worth noting that the equilibrium points are not all ESS since ESS must also possess the ability to resist the error or deviation caused by bounded rationality, i.e., the ability to recover to a stable point after disturbance. Based on the stability theorem of differential equations and the eigenvalues of the Jacobian matrix of differential equations, it can be determined whether the evolutionary system's equilibrium point is stable [63]. When a local equilibrium point satisfies that the determinant of the Jacobian matrix is greater than zero and the trace is less than zero, it is a stable strategy for the evolutionary game [64].

Subsequently, the Jacobian matrix can be obtained as follows [65]:

$$J = \begin{bmatrix} \dfrac{\partial F(x)}{\partial x} & \dfrac{\partial F(x)}{\partial y} \\ \dfrac{\partial F(y)}{\partial x} & \dfrac{\partial F(y)}{\partial y} \end{bmatrix} = \begin{bmatrix} a_{11} & a_{12} \\ a_{21} & a_{22} \end{bmatrix} \tag{11}$$

$$a_{11} = (1 - 2x)[y(\gamma\theta_a + P) + m\theta_a - C_a - J\theta_a] \tag{12}$$

$$a_{12} = x(1 - x)(\gamma\theta_a + P) \tag{13}$$

$$a_{21} = y(1 - y)(\gamma\theta_b + P) \tag{14}$$

$$a_{22} = (1 - 2y)[x(\gamma\theta_b + P) + m\theta_b - C_b - J\theta_b] \tag{15}$$

The determinant and trace of the Jacobian matrix are as follows [66]:

$$\det(J) = a_{11}a_{22} - a_{12}a_{21} \tag{16}$$

$$\mathrm{tr}(J) = a_{11} + a_{22} \tag{17}$$

When $\det(J) > 0$ and $\mathrm{tr}(J) < 0$ are satisfied, it is considered that the fixed point of the local asymptotically stable method corresponds to the evolutionary stable strategy(ESS) [67]. The $\det(J)$ and $\mathrm{tr}(J)$ expression of the five equilibrium points is shown in **Table 3**.

The stability analysis for the five equilibrium points is shown in **Table 4**, among which equilibrium points B(0,0) and D(1,1) are the points of ESS.

**Fig 3** shows the phase diagrams of the evolutionary game. A, B, C, and D constitute the boundary of the evolutionary game domain {(x, y) | x = 0, 1; y = 0, 1}, and the area M enclosed by them is the equilibrium solution domain of the game between the two parties, that is, M = {(x, y) | 0 ≤ x ≤ 1, 0 ≤ y ≤ 1}. Points B(0, 0) and D(1, 1) evolve to be two stable points,

**Table 3. The determinant and the trace of the Jacobian matrix for each equilibrium point.**

| EQUILIBRIUM POINTS | DET(J) | TR(J) |
|---|---|---|
| A(0,1) | $(\gamma\theta_a + P + m\theta_a - C_a - J\theta_a)(-m\theta_b + \theta_b + J\theta_b)$ | $(\gamma + m - J)\theta_a + (J - m)\theta_b + (C_b - C_a) + P$ |
| B(0,0) | $(m\theta_a - C_a - J\theta_a)(m\theta_b - C_b - J\theta_b)$ | $(m - C_a - J)(\theta_a + \theta_b)$ |
| C(1,0) | $(\gamma\theta_b + P + m\theta_b - C_b - J\theta_b)(-m\theta_a + C_a + J\theta_a)$ | $(\gamma + m - J)\theta_b + (J - m)\theta_a + (C_a - C_b) + P$ |
| D(1,1) | $(\gamma\theta_a + P + m\theta_a - C_a - J\theta_a)(\gamma\theta_b + P + m\theta_b - C_b - J\theta_b)$ | $(C_a + J - m)(\theta_a + \theta_b) + (C_a + C_b) - 2P$ |
| E (X*,Y*) | $[(\gamma\theta_a + P + m\theta_a - C_a - J\theta_a)(-m\theta_b + \theta_b + J\theta_b)(\gamma\theta_b + P + m\theta_b - C_b - J\theta_b)(-m\theta_a + C_a + J\theta_a)]/[(\gamma\theta_b + P)(\gamma\theta_a + P)]$ | 0 |

**Table 4. Stability of each equilibrium point.**

| EQUILIBRIUM POINTS | DET(J) | TR(J) | RESULTS |
|---|---|---|---|
| A(0,1) | + | + | unstable |
| B(0,0) | + | + | ESS |
| C(1,0) | + | + | unstable |
| D(1,1) | + | − | ESS |
| E (X*,Y*) | + | 0 | saddle point |

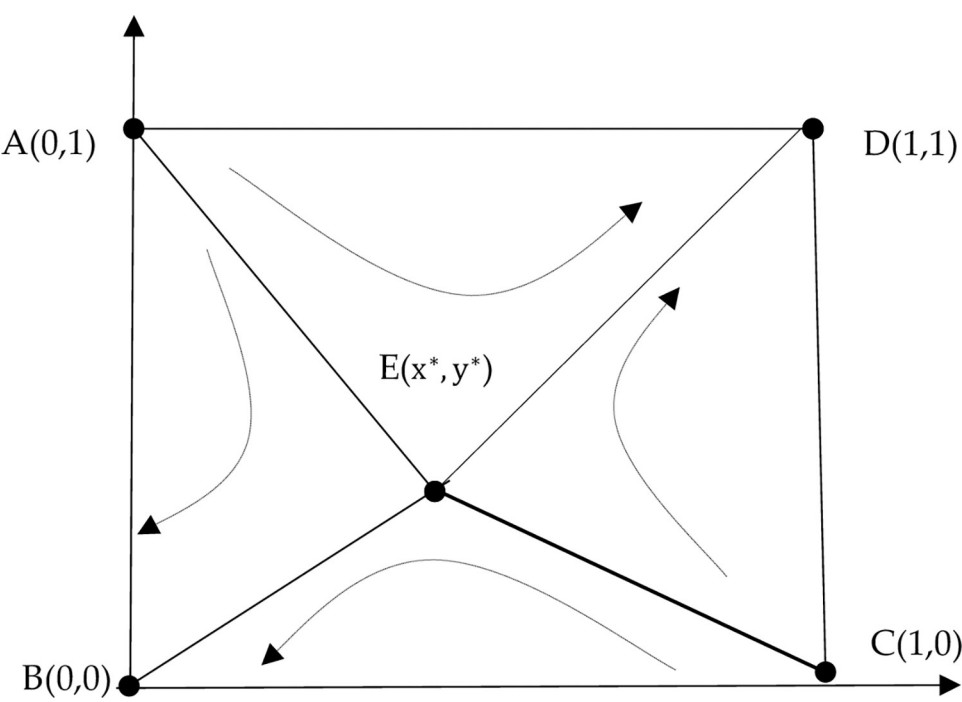

**Fig 3. Phase diagram of the evolutionary game.**

indicating that the replication dynamic curves of players A and B tend to converge to the two points. The discounted AEC divides the graph into two different evolutionary results: "performance" or "breach". When the dynamic curves of the two imitators converge to B(0,0), players A and B will adopt {breach, breach}. When the dynamic curves of the two imitators converge to point D(1,1), Players A and B will adopt {performance, performance}. Point $E(C_b + J\theta_b - m\theta_b/\gamma\theta_b + P, C_a + J\theta_a - m\theta_a/\gamma\theta_a + P)$ is the key point to judge the convergence probability of the two replication dynamic curves to points B and D. If the initial state of both players in the game is near point E, subtle changes change the dynamic evolution results of both players. The final trend of the game participants depends on the comparison between area S1 of region ABCE and area S2 of region AECD. When S2 > S1, both players in the game tend toward the evolutionary result of cooperation. Otherwise, when S2 < S1, the final strategy choice of both players evolves in non-cooperation. The position of the saddle point $E(C_b + J\theta_b - m\theta_b/\gamma\theta_b + P, C_a + J\theta_a - m\theta_a/\gamma\theta_a + P)$ affects the final evolution result.

From the expression of point $E(x^*, y^*)$, it can be seen that the factors that affect the choice of initial real information sharing or non-sharing by both parties in the game include the information cost of sharing initial real information, amount of sharing initial real information, synergy coefficient, reward incentive coefficient, default penalty amount, and the risk coefficient. Changes in these parameters may cause the game results to tend to different balances.

## 4 Numerical simulation of the evolutionary game model

This paper used numerical simulation to further validate the above-constructed evolutionary game model. The above analysis shows that the game's evolution is affected by the position change of the saddle point. *Matlab 22b* is used to simulate the dynamic evolution process of the action selection of each node in the blockchain system under the change of various factors [68]. This simulation enables further discussion of the influence of parameters on the system

evolution results. Through sensitivity analysis, we finally draw corresponding conclusions in chapter 5, and these can provide a more targeted decision-making basis for promoting better development of the system [69].

According to the relevant literature [70], the initial values of each parameter are set as $C_a = C_b = 9$, $\theta_a = \theta_b = 30$, $J = 0.6$, $m = 0.5$, $\gamma = 0.6$, $P = 10$, $L = 0.8$.

## 4.1 The influence of the initial willingness of participants

Under the initial conditions, the position of saddle point E can be calculated as approximately (0.43, 0.43). Through a *Matlab 22b* simulation, the reflection of x and y on the probability of choosing an equilibrium strategy can be obtained.

**Fig 4** shows that (x, y) affects the final strategies of both parties in the game. When the probability x of players A and B choosing to honor the contract is less than 0.43, the initial value of (x, y) falls in the ABCE region. In this case, the initial value converges to (0,0), and players A and B choose the game strategy of {breach, breach} and refuse to share the initial true information of the supply chain. When the probability x of players A and B choosing cooperation is greater than 0.43, the initial value of (x, y) falls in the OECD region, and the initial value converges to (1,1). Now players A and B choose the game strategy of {performance, performance} to share the initial true information about the supply chain. The initial values are set as follows: (0.2, 0.3), (0.6, 0.8), (0.5, 0.5), (0.7, 0.6), (0.1, 0.4), and (0.3, 0.4) to further verify the influence of the cooperation probability of both players on the final game strategy. **Fig 5** depicts the results of this simulation.

**Fig 6** shows that if player A is unwilling to share the initial real information of the AFSC, then $x = 0.2$. If the willingness of player B to share the initial real information of the agricultural supply chain is not strong ($y = 0.2$ to 0.5), then players A and B choose the game strategy of "*breach*" and refuse to share the initial real information of the supply chain. However, if the willingness of player B to share the initial real information of the agricultural supply chain is strong ($y = 0.8$), then players A and B choose the game strategy of {performance, performance} and share the initial real information of the agricultural supply chain.

## 4.2 The influence of information cost of sharing initial real information

Under the premise that the benchmark parameters remain unchanged, an isolated increase in the information cost of sharing initial real information changes the direction in which players A and B finally converge to the equilibrium state, as shown in **Fig 7**. The information costs are set as follows: ($c_1 = 5$, $c_2 = 7$), ($c_1 = 10$, $c_2 = 12$), ($c_1 = 13$, $c_2 = 15$). From **Fig 7**, we can see that with the continuous increase in the information cost of sharing initial real information quality, the evolution result of the system has changed from the original (1,1) to (0,0). When the information cost is higher, the AFSC participants will inevitably demand higher income. If the expected income is not met, it becomes difficult to continue sharing the initial true information about the supply chain. Therefore, higher information costs slow the pace of convergence towards the ideal state for both players.

## 4.3 The influence of reward coefficient

Under the premise that the benchmark parameters remain unchanged, reward coefficients are set as follows: $m = 0.2, 0.6, 0.8, 0.9$. **Fig 8** shows that with the continuous increase in the reward coefficient, the evolution result of the system has changed from the original (0,0) to (1,1). When both sides of the game choose to share initial real information, in other words, the evolutionary stability strategy tends to be {*performance, performance*}. This trend indicates that the supply chain can offer positive incentives to the sharing behavior of initial real

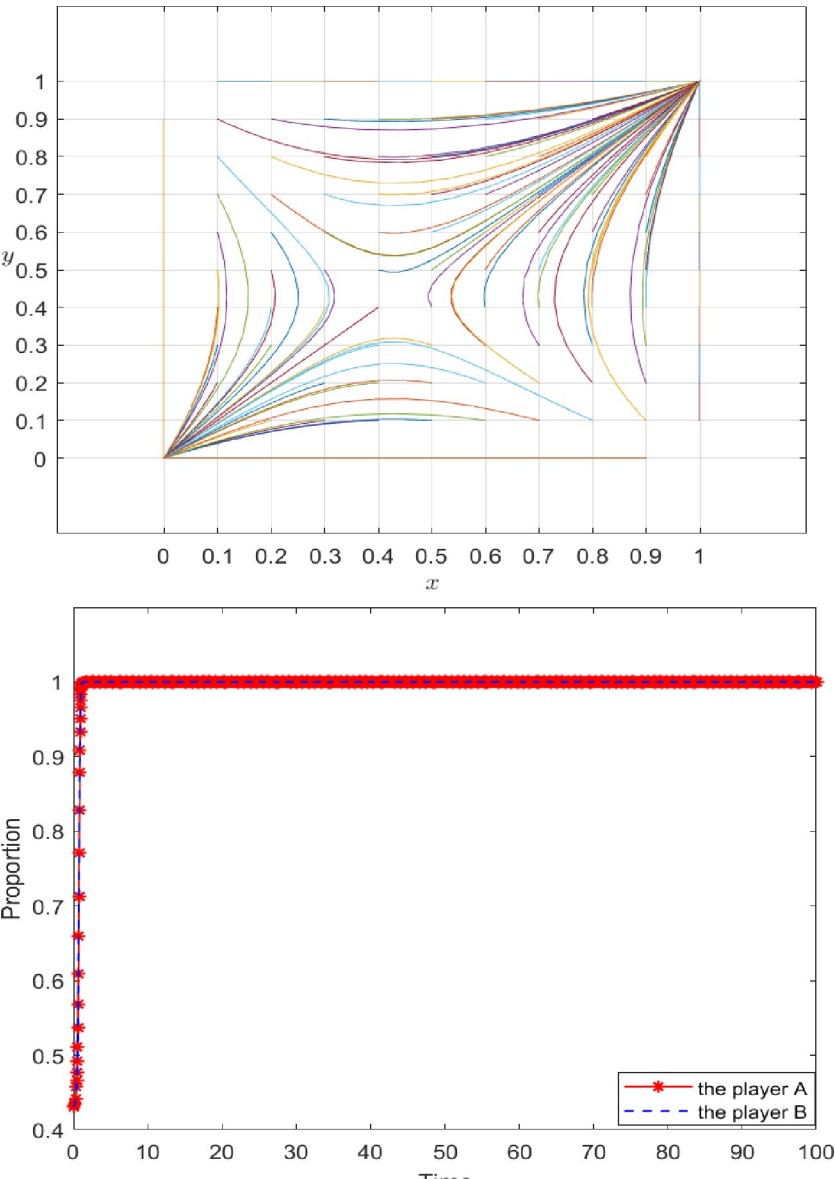

**Fig 4. System evolution results under the initial parameter setting.**

information, and guide participants to carry out initial real information-sharing activities. These actions can help prevent the AFSC participants from wanting to obtain the initial real information of the supply chain through "free-riding" behavior and eventually falling into the "prisoner's dilemma." This kind of incentive includes not only material incentives but also non-material incentives. For example, because the active sharing of initial real information can be recognized, respected, and trusted by other supply chain members, it is conducive to improving the image of participating subjects.

## 4.4 The influence of risk coefficient

Assuming that the benchmark parameters remain unchanged, the risk coefficient is set as follows: $j$ = 0.9, 0.7, 0.4, 0.2. **Fig 9** shows that the smaller the risk coefficient, the probability that

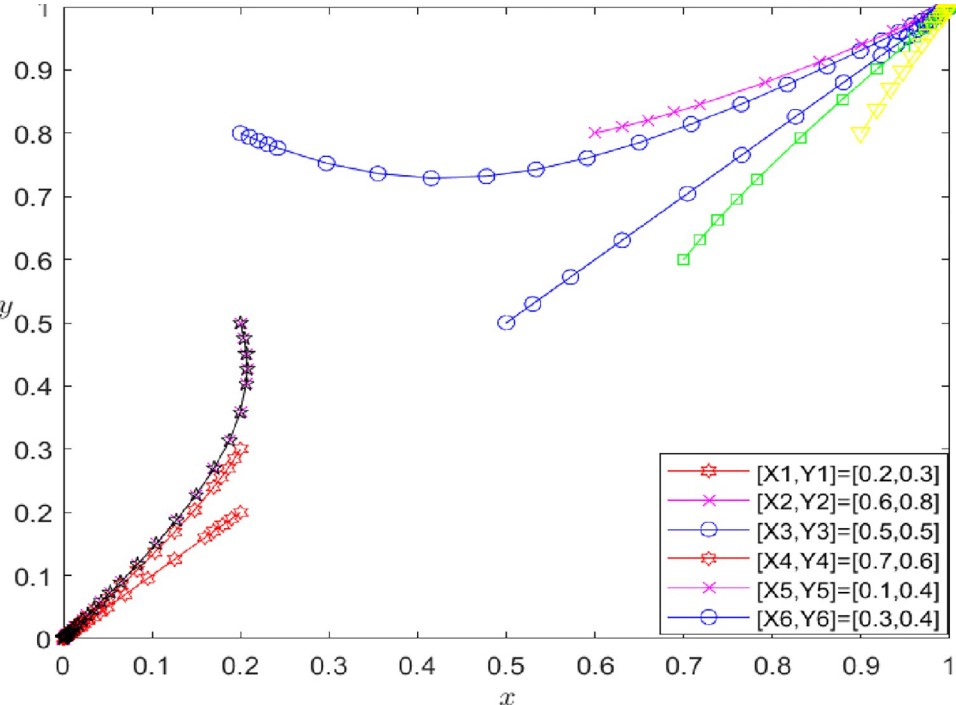

**Fig 5. The impact of (x, y) on evolutionary results.**

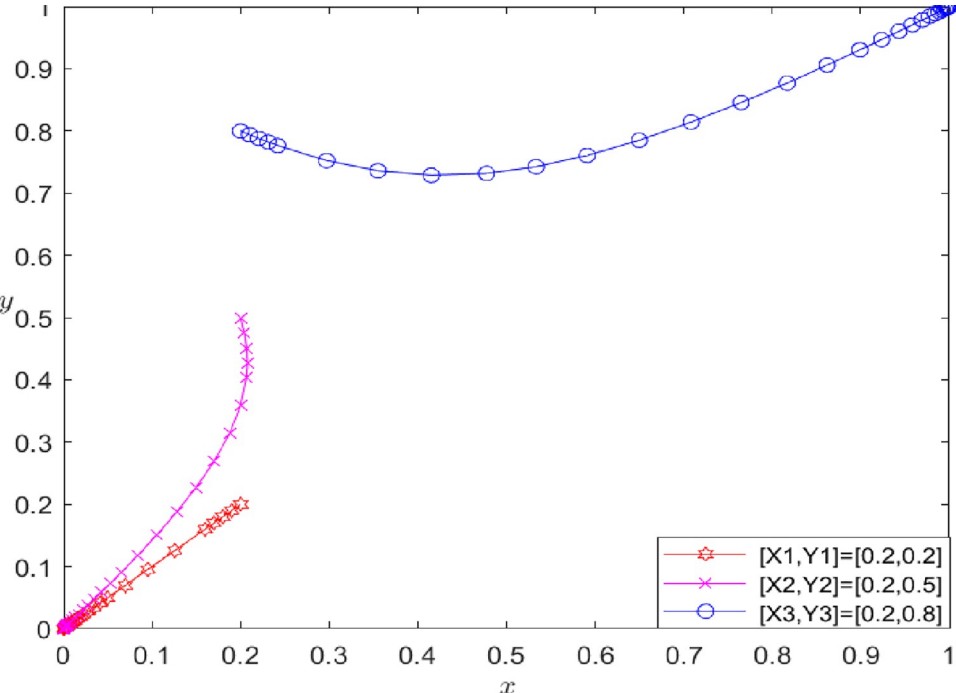

**Fig 6. The impact of the initial willingness of participants on evolutionary results.**

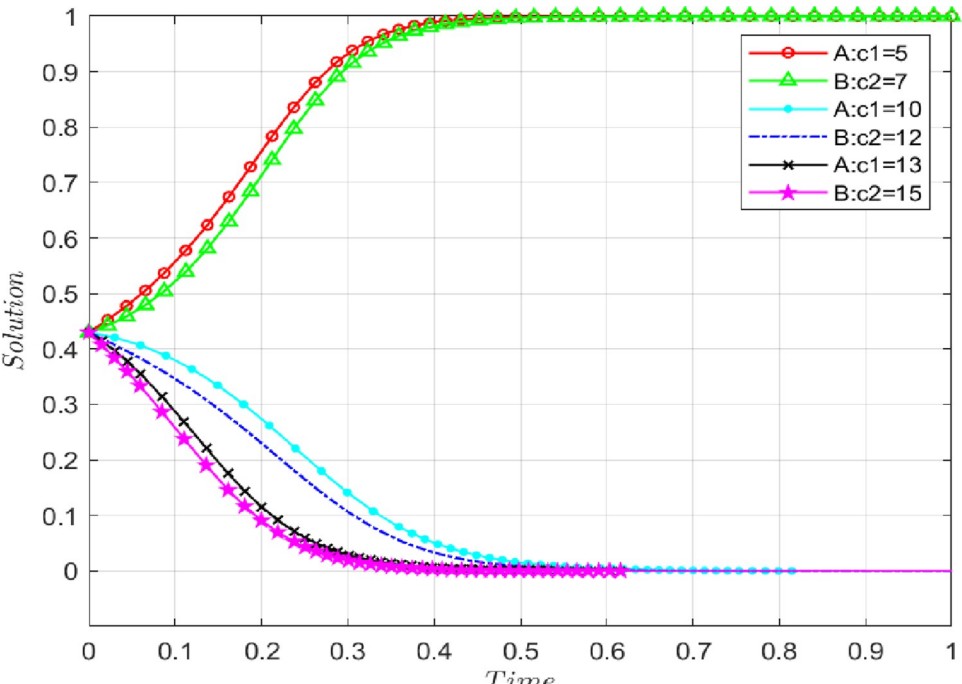

**Fig 7. The impact of sharing initial real information on evolutionary results.**

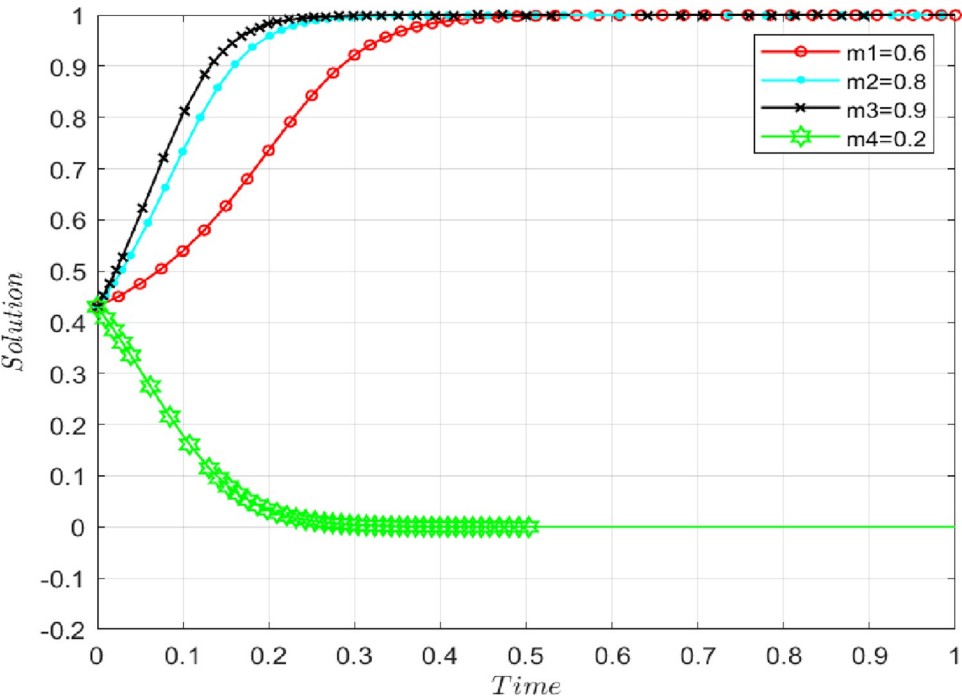

**Fig 8. The impact of the reward coefficient on evolutionary results.**

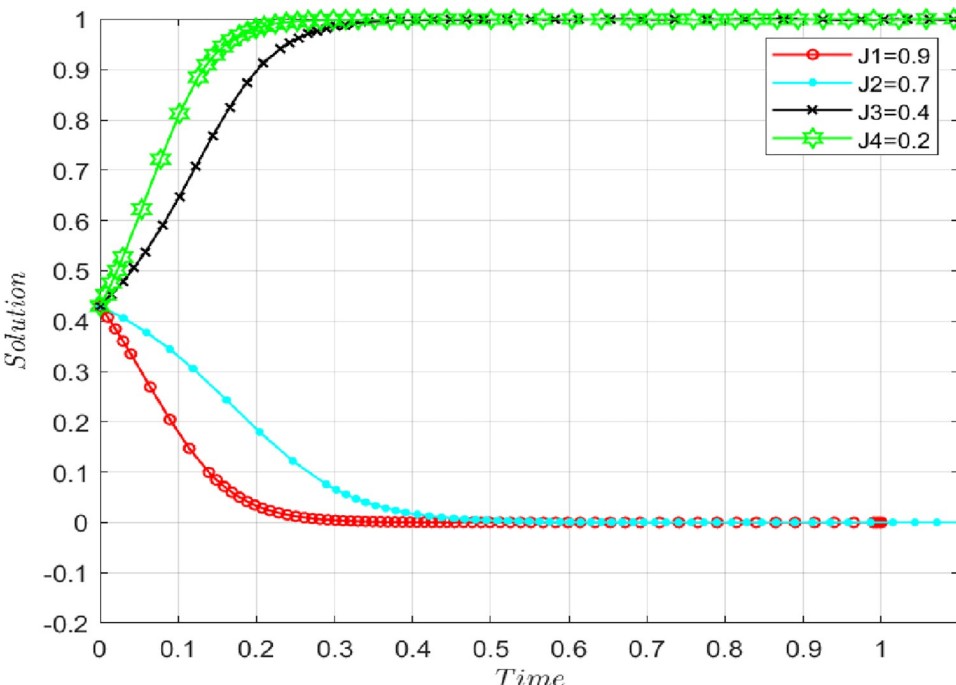

**Fig 9. The impact of risk coefficient on evolutionary results.**

the evolutionary game converges to point D(1,1) increases. That is with the increase in the risk factor if supply-chain participants share larger quantities of true information, their core technologies and trade secrets may be stolen, resulting in losses, so participants stop sharing the initial true information to maximize their interests. When the risk is greater, the trend becomes more pronounced.

## 4.5 The influence analysis of the synergistic effect coefficient

In the case of keeping other parameters unchanged, let the synergistic effect coefficient γ = 0.3, 0.6, 0.8, and 0.9, as shown in **Fig 10**. With the increase in the synergistic effect coefficient, the probability of both players choosing to obey the contract increases. Therefore, the synergistic effect coefficient γ will positively impact the initial true information-sharing behavior. The more apparent synergistic effects lead to a greater probability of initial true information sharing.

## 4.6 The influence of default penalty

Under the premise that the benchmark parameters remain unchanged, an isolated increase occurs in the default penalty when P = 5, 10, 20, and 45. When the default penalty increases, the probability that both sides of the game choose to share the initial real information increases, and the evolutionary game converges at point D (1, 1). The evolutionary stable strategies of both sides of the game tend to choose {performance, performance}. Conversely, the evolutionary stable strategies of both sides of the game tend to choose {breach, breach}, as shown in **Fig 11**.

Therefore, the supply chain can offer positive incentives to the initial true information-sharing behaviors while it punishes "free-rider" behaviors. This strategy encourages supply-chain participants to choose initial real information-sharing behaviors, but the punishment needs to be moderate. If the punishment is too severe, members of the supply chain may not choose to share the initial true information.

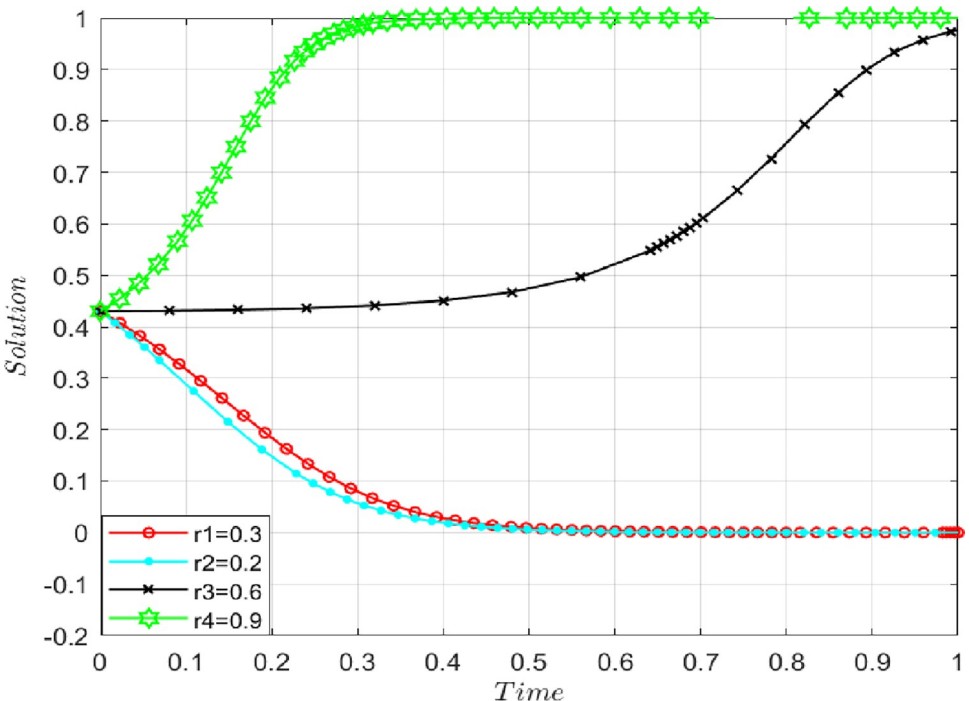

**Fig 10. The impact of the synergistic effect coefficient on evolutionary results.**

## 4.7 The influence of amount of information sharing

The amount of information sharing can significantly impact the initial real information sharing. Under the premise that the benchmark parameters remain unchanged, an isolated

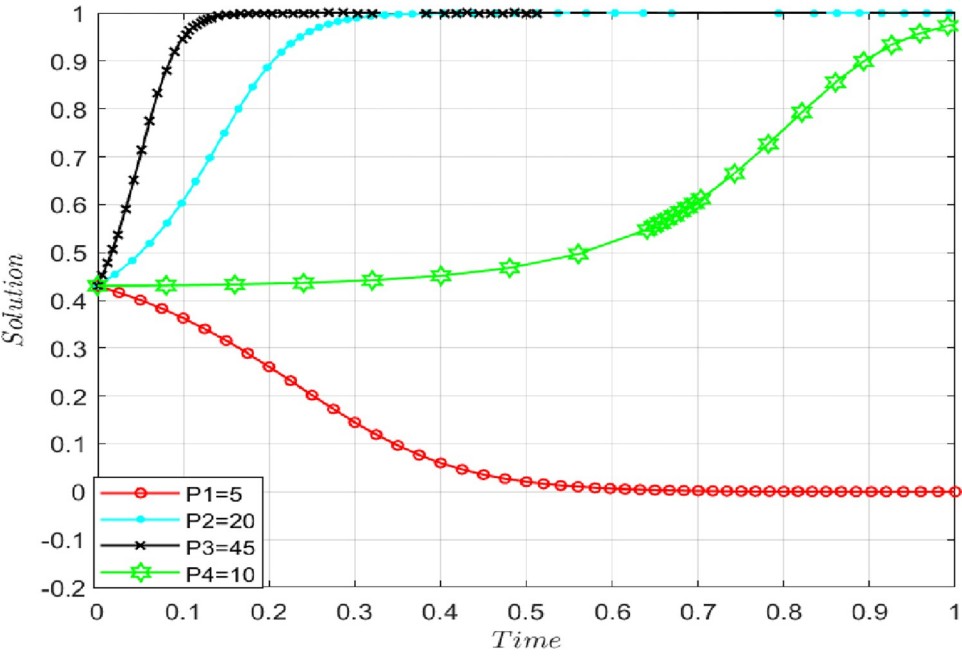

**Fig 11. The impact of default penalty on evolutionary results.**

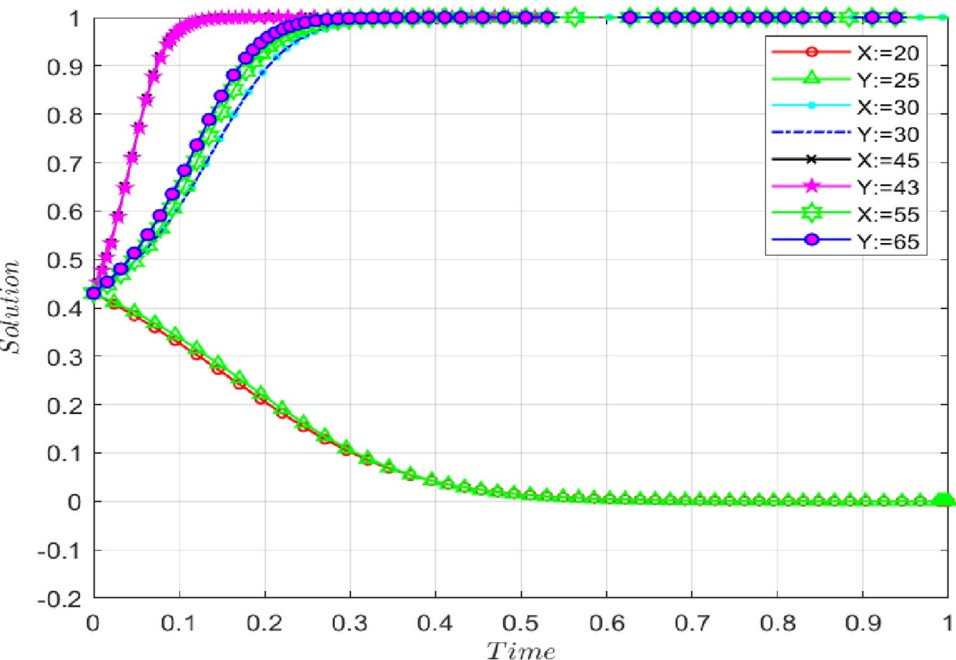

**Fig 12. The impact of the amount of information sharing on evolutionary results.**

increase in the amount of information sharing changes the direction in which players A and B finally converge to the equilibrium state, as shown in **Fig 12.** The amount of information sharing is set as follows: ($\theta_a = 20$, $\theta_b = 22$), ($\theta_a = 30$, $\theta_b = 27$), ($\theta_a = 45$, $\theta_b = 43$), and ($\theta_a = 55$, $\theta_b = 58$). When the amount of information sharing increases, the probability that both sides of the game choose to share the initial real information increases. As shown in **Fig 12**, the lower amount of information sharing slows the convergence pace for both players A and B towards the ideal state.

## 5. Conclusions

### 5.1 Conclusion of evolution results

In the AFSC, information sharing under blockchain technology has become increasingly critical in driving sustainable development. Yet, the question of how the sharing of the initial real information of each node enterprise in the AFSC in the blockchain is influenced and evolved has been inadequately answered. This study investigates the initial real information sharing in the AFSC. A dynamic game model of the agri-food supply chain participants is constructed to explore the implementation process of the two parties' evolutionary strategies.

By establishing the interest matrix, the replication dynamic equation can be obtained for both sides of the game. According to the replication dynamic equation, the Jacobian matrix and the sign of the determinant and trace can be obtained. This information can then be used to find the five equilibrium points of the game model and their stable states.

Our main results are as follows: First, we prove that the equilibrium point where players A and B simultaneously choose the "*performance, performance*" strategy or "*breach, breach*" strategy is the ESS stable point, which has long-term dynamic stability. Second, we demonstrate that the stability of the chosen strategy of both sides in the game is related to the parameter variation of the replication dynamic equation. Through *Matlab 2022b* simulation verification and analysis results, it is found that the greater the risk coefficient of using blockchain technology,

the higher the information cost of sharing initial real information. Thus, the agricultural supply chain participants are more inclined to breach the contract and not abide by the rules of the blockchain system to share the initial true information. When the reward incentive coefficient, synergy coefficient, default penalty amount, and amount of sharing initial real information are larger, the participants are more inclined to keep the agreement. They abide by the rules of the blockchain system, actively share the initial true information, and strengthen the sharing of the initial true information through the incentive and punishment mechanism.

## 5.2 Suggestions and prospects

Based on the results above, we propose the following insights for the leading enterprise in the agricultural supply chain when formulating a strategy for promoting sharing information. First, strict access and constraint mechanisms are implemented to protect the authenticity of the initial information. The goal is to form the restriction among the supply-chain participants, prevent opportunism and free-riding behavior, such as obtaining extra income by lying and cheating, and increase the amount of sharing initial real information of the system nodes. Second, an effective benefit-distribution mechanism is formulated to promote full cooperation and effective communication among the participants of each node enterprise to achieve synergistic benefits. The cooperation of upstream and downstream enterprises to fulfill information sharing can promote the sustainable development and operation of the agricultural supply chain, promote the establishment of a rural industrial system in rural areas, and maintain the comprehensive revitalization of social harmony and stability. Third, the breach of the initial information, sharing of false information, and undermining of the sharing mechanism established by the blockchain system are punished. The whole process of notification is carried out on the blockchain system chain, which affects the future financing and transaction of enterprises and the loss of reputation. Following the blockchain system node that is sharing real information involved in the main body in the form of currency encrypted blocks can create compensation incentives and transaction costs, improve the credit rating through rewards and punishments constraints and incentives to reduce certain risks, strengthen the true information, and improve the quantity of information sharing. Finally, the government should give full play to the synergistic effect of corporate social responsibility, actively encourage enterprises to produce and sell agri-food to fulfill corporate social responsibility, and promote the sustainable development of the economy and society.

## 5.3 Research limitations

We recognize and acknowledge the existence of several important limitations in this paper, which also open doors to future research. First, because of the complexity of the model, this study did not consider all the factors affecting the sharing of the initial real information of the AFSC in the blockchain. Only some significant and acknowledged influencing factors were considered in the evolutionary game. Second, to make the factors considered in the research and analysis more comprehensive, more subjects(for example, government and logistics enterprises, etc) need to participate in the game and analyze the factors that affect their decision-making.

Along with the findings proposed in this paper, several aspects can be pursued as future research. First, a project that extends the current model setting by incorporating government and logistics enterprises to construct a tripartite evolutionary game model is worth investigating. Second, some significant and acknowledged influencing factors(for example, the government's fiscal constraints and logistics enterprise, etc) were considered in the evolutionary game.

## Author Contributions

**Conceptualization:** Weixia Yang, Congli Xie, Lindong Ma.

**Data curation:** Weixia Yang, Congli Xie, Lindong Ma.

**Formal analysis:** Weixia Yang, Congli Xie.

**Funding acquisition:** Weixia Yang, Congli Xie.

**Investigation:** Weixia Yang, Lindong Ma.

**Methodology:** Weixia Yang, Congli Xie, Lindong Ma.

**Project administration:** Weixia Yang, Congli Xie, Lindong Ma.

**Resources:** Weixia Yang, Congli Xie.

**Software:** Weixia Yang, Congli Xie.

**Supervision:** Weixia Yang, Congli Xie, Lindong Ma.

**Validation:** Weixia Yang, Congli Xie, Lindong Ma.

**Visualization:** Weixia Yang, Congli Xie.

**Writing – original draft:** Weixia Yang, Congli Xie, Lindong Ma.

**Writing – review & editing:** Weixia Yang, Congli Xie, Lindong Ma.

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
