## [Decision Letter · Decision Letter 0]

3 Apr 2023

PONE-D-23-02505Dose Blockchain-Based Agri-food Supply Chain Guarantee the Initial Information Authenticity? An Evolutionary Game PerspectivePLOS ONE

Dear Dr. Ma,

Thank you for submitting your manuscript to PLOS ONE. After careful consideration, we feel that it has merit but does not fully meet PLOS ONE’s publication criteria as it currently stands. Therefore, we invite you to submit a revised version of the manuscript that addresses the points raised during the review process. The reviewers recommend reconsideration of your manuscript following major revision. I invite you to resubmit your manuscript after addressing the comments below.Please submit your revised manuscript by May 18 2023 11:59PM. If you will need more time than this to complete your revisions, please reply to this message or contact the journal office at plosone@plos.org. Please include the following items when submitting your revised manuscript:A rebuttal letter that responds to each point raised by the academic editor and reviewer(s). You should upload this letter as a separate file labeled 'Response to Reviewers'.A marked-up copy of your manuscript that highlights changes made to the original version. You should upload this as a separate file labeled 'Revised Manuscript with Track Changes'.An unmarked version of your revised paper without tracked changes. You should upload this as a separate file labeled 'Manuscript'.

We look forward to receiving your revised manuscript.

Kind regards,

Vijay Kumar

Academic Editor

PLOS ONE

Reviewers' comments:

Reviewer's Responses to Questions

**Comments to the Author**

1. Is the manuscript technically sound, and do the data support the conclusions?

Reviewer #1: Yes

Reviewer #2: Yes

Reviewer #3: Yes

2. Has the statistical analysis been performed appropriately and rigorously? 

Reviewer #1: Yes

Reviewer #2: Yes

Reviewer #3: Yes

3. Have the authors made all data underlying the findings in their manuscript fully available?

Reviewer #1: Yes

Reviewer #2: Yes

Reviewer #3: No

4. Is the manuscript presented in an intelligible fashion and written in standard English?

Reviewer #1: Yes

Reviewer #2: Yes

Reviewer #3: Yes

5. Review Comments to the Author

Reviewer #1: congratulate you on the rigor of your reasoning. Your argumentative logic is particularly robust. The mathematical demonstration is of high quality. The different scenarios are plausible.

However, you avoid an issue that I think is important to mention in the risk management section. You say that there is no transparency in the supply chain of the food industry... But you don't say why? Your article would benefit from stating the possible reasons for this situation. At this stage, I don't see what would be the reasons that could motivate actors to use the blobkchain unless they are forced to do so...Sometimes secrecy is the very condition of value creation.... What do you have to say about this?

Reviewer #2: This study investigates the game theoretic setting in the agri-food supply chain field by analyzing the strategies of game players and their conditions, with the overall purpose to improve this supply chain network. Overall, the paper offers interesting insights to understanding this issue.

But some major and minor revisions are needed to improve the literature review, theoretical background of the approach, and modeling and simulation.

The following suggestions should help to improve the paper.

Major:

1. Introduction.

1.1. In line 84, add the key contributions of your study, e.g. the model helps to solve or understand what? Implications for the players in the supply chain and logistics in agriculture? Lessons learnt from the study. A few brief lines is enough.

2. Literature review.

2.1. The literature review is weak. About the blockchain technologies in agriculture, in the section 2.1 clarify on the sustainability and governance impacts. Also, start the section 2.2 saying that the games can be cooperative and non-cooperative and mention where your proposed model stands in. Also, highlight if your game is extensive or normal. Also, clarify who the public sector and who the private sector player is in the literature, since the game theory models much depend on this. There are a lot of studies on this issue in the public-private partnership literature, which can be briefly added into the section 2.2. The following studies should be added to improve these points:

- "Climate-smart agriculture using intelligent techniques, blockchain and Internet of Things: Concepts, challenges, and opportunities" in TETT 2022 journal.

- "A review of the use of game theory in project management" in JME 2022 journal.

- "Analysis on the stability and evolutionary trend of the symbiosis system in the supply chain of fresh agricultural products" in PO 2020 journal.

- "A meta-analysis of the public-private partnership literature reviews: exploring the identity of the field" in IJSPM 2022 journal.

3. Model, simulations, and results.

3.1. Is your model cooperative or non-cooperative? Please clarify this in the introduction to your model in the section 3.1.

3.2. Also how the equilibrium can be reached in the matrix in Table 2. Please clarify.

3.3. In the section 3.3, define the concept of "stability", even if it leads to robust points to account for minor disturbances, but what is the stability in your study?

3.4. In the section 4 it is not clear what are your input variables/data into the simulation. In the beginning of the section, in line 366-371 clarify this.

4. Conclusions.

4.1. The results are clear and the findings are interesting.

4.2. Another future research can be to study downstream and upstream strategy players separately using such models as the Shapley or Rubinstein game models.

Minor:

1. Line 24 - remove the unnecessary full stop.

2. Line 25 - given a full name for the abbreviation.

3. The case is on Chine, so change the title to include "China" accordingly.

4. There can be other grammar issues to be fixed.

Good luck.

Reviewer #3: 1.The paper also has many writing errors and disfluencies, such as: Line 24：“agri-food” should be capitalized as “Agri-food”; Line 146 “when” should be capitalized as “When”.

2. The literature is not comprehensive enough and limited in amount and without a summary of the latest research and to present research gaps, it is recommended to enrich the content of the literature review.

3.The variable definition in Table 1. does not match the case of the actual variable in the paper, for example, “X/Y ” should be changed to “x/y”, “M” should be “m”.

4.Figure 1. missing "P" in the inequality sign of (c).

5.Figure 4. has two diagrams without separate labels.

6.Line 200 F(x) and Line 252 F(y) should be expressed as F1(x,y) and F2(x,y), respectively.

7.The overall volume of the paper is slightly less, and it is suggested to add more contents, such as using the three-way evolutionary game approach to study the game strategy selection problem of three groups.

8.There are many studies on evolutionary games that choose tripartite subjects, such as Zheng, Y et al. (2023) and Wang, F et al. (2022) which addresses blockchain traceability of agricultural supply chains and quality and safety of agricultural products, respectively.

References

[1] Blockchain Traceability Adoption in Agricultural Supply Chain Coordination: An Evolutionary Game Analysis. Agriculture 2023, 13, 184. https://doi.org/10.3390/ agriculture13010184

[2] Evolutionary Game Analysis of the Quality of Agricultural Products in Supply Chain. Agriculture 2022, 12, 1575. https://doi.org/10.3390/ agriculture12101575

6. PLOS authors have the option to publish the peer review history of their article (what does this mean?). If published, this will include your full peer review and any attached files.

Reviewer #1: **Yes: **Nicolas Merveille

Reviewer #2: No

Reviewer #3: No

---

## [Author Response · Author response to Decision Letter 0]

18 Apr 2023

Dear Reviewer,

 We greatly appreciated your constructive and critical remarks on the manuscript. The comments and suggestions have helped improve the contributions of the paper. Thank you sincerely.

Following your recommendations and suggestions, we have made an extensive revision of the manuscript. The paper is also thoroughly proofread to rectify any grammatical and typographical errors. We also provide detailed responses to your comments. For easy identification, major revisions are marked in blue in the manuscript. Referee comments are in italics in this response. Quotes from the paper are in blue in this report. We hope the revised manuscript meets the publication standards.

Thanks again for your recommendations. For more detailed responses, please check the attached file.

Sincerely,

Corresponding Author

---

## [Decision Letter · Decision Letter 1]

25 May 2023

Dose Blockchain-Based Agri-food Supply Chain Guarantee the Initial Information Authenticity? An Evolutionary Game Perspective

PONE-D-23-02505R1

Dear Dr. Ma,

We’re pleased to inform you that your manuscript has been judged scientifically suitable for publication and will be formally accepted for publication once it meets all outstanding technical requirements.

Kind regards,

Vijay Kumar

Academic Editor

PLOS ONE

Additional Editor Comments (optional):

One of the reviewers has highlighted that the manuscript has typographical and language issues that should be addressed, such as: Line 84 “Theoretical level.” is not a complete sentence and thus should be deleted. The authors must proofread the manuscript carefully before final publication.

Reviewers' comments:

Reviewer's Responses to Questions

**Comments to the Author**

1. If the authors have adequately addressed your comments raised in a previous round of review and you feel that this manuscript is now acceptable for publication, you may indicate that here to bypass the “Comments to the Author” section, enter your conflict of interest statement in the “Confidential to Editor” section, and submit your "Accept" recommendation.

Reviewer #2: All comments have been addressed

Reviewer #3: (No Response)

2. Is the manuscript technically sound, and do the data support the conclusions?

Reviewer #2: Yes

Reviewer #3: (No Response)

3. Has the statistical analysis been performed appropriately and rigorously? 

Reviewer #2: Yes

Reviewer #3: (No Response)

4. Have the authors made all data underlying the findings in their manuscript fully available?

Reviewer #2: Yes

Reviewer #3: (No Response)

5. Is the manuscript presented in an intelligible fashion and written in standard English?

Reviewer #2: Yes

Reviewer #3: (No Response)

6. Review Comments to the Author

Reviewer #2: I have reviewed the revised manuscript again. The authors have addressed all the major comments raised in the earlier version of the paper. I wish good luck to the authors and hope the paper will be interesting to the journal readership.

Reviewer #3: (No Response)

7. PLOS authors have the option to publish the peer review history of their article (what does this mean?). If published, this will include your full peer review and any attached files.

Reviewer #2: No

Reviewer #3: No

---

## [Editor Report · Acceptance letter]

15 Jun 2023

PONE-D-23-02505R1 

Dose Blockchain-Based Agri-food Supply Chain Guarantee the Initial Information Authenticity? An Evolutionary Game Perspective 

Dear Dr. Ma:

I'm pleased to inform you that your manuscript has been deemed suitable for publication in PLOS ONE. Congratulations! Your manuscript is now with our production department. 

Kind regards, 

on behalf of

Dr. Vijay Kumar 

Academic Editor

PLOS ONE